# Empowerment and Knowledge as Determinants for Quality of Life: A Contribution to a Better Type 2 Diabetes Self-Management

**DOI:** 10.3390/ijerph20054544

**Published:** 2023-03-03

**Authors:** Pedro L. Ferreira, Carminda Morais, Rui Pimenta, Inês Ribeiro, Isabel Amorim, Sandra Maria Alves

**Affiliations:** 1Centre for Health Studies and Research, University of Coimbra, 3004-512 Coimbra, Portugal; 2Faculty of Economics, University of Coimbra, 3004-512 Coimbra, Portugal; 3Superior School of Health, Polytechnic of Viana do Castelo, 4900-314 Viana do Castelo, Portugal; 4School of Health, Polytechnic of Porto, 4200-072 Porto, Portugal

**Keywords:** QoL determinants, type 2 diabetes, chronic disease self-management, Diabetes Empowerment Scale, Diabetes Knowledge Test, EQ-5D-5L

## Abstract

The purpose of this study was to assess how knowledge and empowerment impact the quality of life (QoL) of a person with type 2 diabetes, leading to better communication and disease management. We conducted a descriptive and observational study of individuals with type 2 diabetes. The Diabetes Empowerment Scale-Short Form (DES-SF), Diabetes Knowledge Test (DKT), and EQ-5D-5L were used, in addition to sociodemographic and clinical characteristics. Evaluating the variability in the DES-SF and DKT in relation to the EQ-5D-5L and identifying possible sociodemographic and clinical determinants were conducted using univariate analyses followed by a multiple linear regression model to test whether the factors significantly predicted QoL. A total of 763 individuals were included in the final sample. Patients aged 65 years or older had lower QoL scores, as well as patients who lived alone, had less than 12 years of education, and experienced complications. The insulin-treated group showed higher scores in DKT than the non-insulin-treated group. It was also found that being male, being under 65 years of age, having no complications present, and having higher levels of knowledge and empowerment predicted higher QoL. Our results show that DKT and DES are still determinants of QoL, even after adjusting for sociodemographic and clinical characteristics. Therefore, literacy and empowerment are important for the improvement of the QoL of people with diabetes, by enabling them to manage their health conditions. New clinical practices focused on education, increasing patients’ knowledge, and empowerment may contribute to better health outcomes.

## 1. Introduction

Diabetes mellitus (DM) is a metabolic disease characterized by hyperglycemia resulting from defects in insulin secretion and/or its action [1,2,3]. The incidence and prevalence of diabetes present geographically variable distributions [4,5] and are estimated to be continuously increasing across all continents, involving significant, non-negligible costs [4,6].

According to the Organization for Economic Co-operation and Development (OECD), in 2019, the estimated standardized prevalence of diagnosed diabetes in the European population aged 15 years or more was over 7%. Compared to this European figure, Portugal showed a rate of 9%, one of the highest prevalence rates in Europe [5]. However, between 2009 and 2019, due to an evidently better performance in primary health care, the age and sex-standardized rates of avoidable hospital admissions for adults with diabetes was 56 per 100,000 inhabitants, the fourth-best European score.

Type 2 DM, like some other chronic non-communicable diseases, is asymptomatic, representing an increased risk of long-term complications. In this context, disease management is a constant challenge. In fact, coping with a chronic disease, such as diabetes, involves dealing with specific physical aspects of the disease. Dietary self-regulation (diet, food selection, proportions), monitoring blood glucose levels, physical exercise, drug therapy (dose, frequency, timing), foot care, and stress control are fundamental aspects of type 2 diabetes self-care management [4,7,8]. This involves psychosocial changes and limitations, as well as associated concerns, such as adherence to therapeutic regimes and uncertainty about the future, resulting in a series of losses leading to changes in independence and decreased well-being and QoL [9]. 

The prevention of long-term complications is also associated with metabolic control [4,9]. However, effective diabetic management cannot be achieved only under the supervision of health professionals [4,6]. The individual with diabetes plays a central role in the effectiveness and self-management of his/her healthcare process.

The management of diabetes implies an adequate level of knowledge in order to develop critical awareness about the necessity of changes regarding the therapeutic process [10], thus promoting its effective adoption [4,9]. It requires not only the acquisition and deepening of specific knowledge in the area of diabetes but also the ability of people to set their own goals and objectives. 

To ensure that people with diabetes and their families acquire “skills for health”, in the terms recommended by the World Health Organization (WHO), it is necessary to include them in the center of the care management process [11]. This implies the rupture of paternalistic healthcare models, where people are seen as mere passive recipients [12]. There is an urgent need to (re)orient clinical practices based on collaborative approaches, to empower people with diabetes to control lifestyle habits and health behaviors and make more appropriate decisions for their condition [13].

In this context, clinical practices must assume an interactive and multidirectional approach based on the values, principles, culture, and individual experiences of people with type 2 diabetes and their families. More than adapting people to their disease, it is important to take people as care partners, to enhance their endogenous (such as locus of control, motivation, and personality characteristics) and exogenous resources (like access to health information and equipment and economic aspects), and to cope with the management of this chronic and complex disease [13,14]. 

Thus, collaborative psychotherapeutic and motivational approaches oriented to self-management form the basis of empowerment [6]. 

The identification of the level of knowledge about type 2 DM (including diet, lifestyle, therapeutic management, and complications [15]), empowerment, and self-management are central concepts in this research.

Empowerment is considered a social process in which people take ownership of their own lives through interactions with others, producing critical thinking about reality and promoting the construction of social and personal capacities [16]. In line with the WHO, empowerment is characterized as the personal and collective ability to contribute to the construction of policies and services more oriented to their needs and potentialities [11,17]. Self-management is defined as a person’s strategies for controlling the disease, promoting health, and living well with the disease [18]. 

Several studies revealed that empowerment can be associated with an improvement in the clinical and non-clinical outcomes of people with type 2 diabetes, such as HbA1c, behavioral changes, health literacy, psychological status, self-care, and control [19,20,21,22,23]. Therefore, to increase empowerment among these patients, it is important that they have sufficient knowledge about the disease and their health status [23]. A systematic review of the factors associated with glycemic control showed that patients’ knowledge regarding diet enhanced their dietary self-regulation and led to informed decision-making adapted to their health condition [24]. Possessing knowledge about the disease has been found to be conducive to the successful self-management of type 2 diabetes through the development of activities, skills, and self-care attitudes [21].

Therefore, when aiming for effective metabolic control, QoL, and subjective well-being [25], these conceptions confirm the possibilities of combining personal and collective strategies [6] involving people with diabetes, their families, the community environment, and health professionals.

In this sense, it is important to proceed with the research supporting clinical practice on the QoL of people with type 2 diabetes and its relationship with empowerment and knowledge, with a focus on more person- and family-centered care models. 

Based on these assumptions, this study aimed to assess how knowledge and empowerment impacted the QoL of a person with type 2 diabetes, with the purpose of contributing to an improvement in self-management processes. It also aimed to explore possible sociodemographic and clinical factors determining QoL.

## 2. Materials and Methods

### 2.1. Study Design

We carried out a descriptive and observational study in four health institutions in the North Region of Portugal. The research project was approved by the Ethical Committee of the Northern Regional Health Authority (62/2018) responsible for all health units where data were collected.

### 2.2. Sample

The population under study was composed of individuals with type 2 diabetes, followed in external consultations of four health institutions. We estimated a total of 85,820 elements, which was the estimated total number of patients with diabetes covered by the four institutions. The sample size was based on a confidence level of 95% and a maximum sampling error of 4%. We obtained a sample of 763 persons who met the following inclusion criteria: a diagnosis of type 2 diabetes over one year, enrolled in multidisciplinary consultation for at least three months, and older than 18 years of age. The recruitment strategy was sequential. Each participant was guaranteed anonymity and confidentiality. Free prior informed written consent was always requested. The questionnaires were self-completed and privacy was safeguarded through the use of separate rooms. The rate of participation was around 90%.

### 2.3. Measurement Instruments

To collect the necessary information for this study, we applied the Portuguese versions of the following measurement instruments: the Diabetes Empowerment Scale-Short Form (DES-SF), the Diabetes Knowledge Test (DKT), and the EQ-5D-5L. In addition, we also collected some sociodemographic variables (sex, age, education, and living alone) and clinical characteristics of the participants (body mass index—BMI, glycated hemoglobin—HbA1c, treatment with insulin, and time of diabetes diagnosis). The following sections provide brief descriptions of each questionnaire.

#### 2.3.1. Diabetes Empowerment Scale-Short Form (DES-SF)

It was developed by Anderson et al. in 2000 and aimed to measure psychological self-efficacy in people with diabetes. The original DES consists of 37 items divided into 8 dimensions: assessing the need for change; developing a plan; overcoming barriers; asking for support; supporting oneself; coping with emotion; motivation; and making appropriate choices in diabetes care, according to priorities and circumstances. The number of items was reduced to 28, encompassing three subscales: (i) assessment of dissatisfaction and readiness to change; (ii) psychosocial management of diabetes; and (iii) goal setting and achievement [26].

To create the DES-SF, the authors choose the item with the highest correlation in each domain of the original scale, resulting in a questionnaire composed of eight items [26]. This measurement instrument presents five response options, from 1 (totally disagree) to 5 (totally agree). The final score was calculated using the average of the scores of the 8 items, in which higher scores indicated better psychosocial self-efficacy. In a sample of 229 participants, the DES-SF proved to have good reliability, with a Cronbach’s alpha of 0.84. Content validity was verified with questionnaire scores and HbA1c levels varied positively after a six-week educational program based on patients’ problems [27]. In this study, we calculated the final score by converting it into a scale from 0 to 100.

In Portugal, Aveiro et al. (2015) used the DES-SF in a sample of the Portuguese population that was composed of 81 people with type 2 diabetes, and this measure demonstrated good internal consistency (Cronbach’s alpha of 0.87) and stability over time (test-retest correlation coefficient of 0.33). The construct validity showed statistically significant differences in the correlation between DES-SF scores and HbA1c levels (r = −0.114) [28].

#### 2.3.2. Diabetes Knowledge Test (DKT)

This measurement instrument was developed by the Michigan Diabetes Research Training Center in the 1990s, and it was created to assess the overall knowledge that a person with diabetes has about the disease. It is composed of two subscales: the first is composed of 14 items and is appropriate for adults with type 1 or type 2 diabetes; the second is composed of nine items specifically for people with diabetes treated with insulin. The questionnaire takes about 15 min to complete [29]. The scoring of this instrument is performed according to the number of right answers.

Fitzgerald et al. (1998) verified the reliability of this questionnaire, which showed Cronbach’s alpha values greater than 0.70 in both subscales. The authors formulated several hypotheses to assess construct validity and concluded that there were statistically significantly higher DKT scores among people with type 1 diabetes, people with a higher level of education, and people who had participated in a diabetes therapy education program [29]. This instrument has been widely used in studies in Portugal, and an association between knowledge about the disease and the ability to control type 2 diabetes has already been demonstrated [15,30].

#### 2.3.3. EQ-5D-5L

This instrument was developed in 2011 by the EuroQol group, with the purpose of creating an instrument capable of measuring health-related quality of life. It is composed of five dimensions (mobility, self-care, usual activities, pain/discomfort, and anxiety/depression), and it offers five response options, ranging from 1 (no problems) to 5 (extreme problems), representing five levels of severity for each dimension [31]. Through a mathematical algorithm, it is possible to convert the response values into a personal quality of life index, ranging from 0 (death) to 1 (perfect health), and allow for the calculation of health-related quality of life that corresponds with the person’s health status [32]. At the end of the questionnaire, there is a visual analog scale (EQ-VAS), where the person identifies his/her current health status, from 0 (worst imaginable health status) to 100 (best imaginable health status) [33].

In 2013, the Portuguese version of the EQ-5D was adapted and validated. This questionnaire proved to be well accepted and reliable, with a Cronbach’s alpha of 0.71 across the five dimensions and an intraclass correlation coefficient of 0.86 for the EQ-VAS. This instrument was also revealed to be feasible, having demonstrated construct and criterion validity with the 36-item Short-Form Health Survey [33]. In 2019, Ferreira et al. calculated the EQ-5D-5L health state preferences of the Portuguese population. The societal value set varied between −0.603 and 1 [34].

### 2.4. Statistical Analysis

Descriptive analyses were conducted to characterize the sample and as a complement to inferential statistics. The Spearman coefficient was used to obtain correlations between the DES-SF, DKT, and EQ-5D-5L (after Kolmogorov–Smirnov normality tests were applied). Cronbach’s alpha coefficients were calculated to assess the reliability of the instruments. The observed Cronbach’s alpha coefficients for the instruments were 0.80 for the DES-SF, 0.81 for the DKT, and 0.81 for the EQ-5D-5L, indicating good internal consistency [35].

The variability of the DES-SF and DKT on the EQ-5D-5L, and the identification of possible sociodemographic and clinical determinants, were evaluated using univariate analyses via independent *t*-tests, followed by a linear regression model using the backward method of variable selection (with a *p*-value > 0.1 as a threshold for variable elimination). Residual analyses were conducted to assess normality, linearity, and homoscedasticity. In addition, to further evaluate homoscedasticity, the Breusch–Pagan test was used as described in Gujarati [36], and collinearity was evaluated using the variance inflation factor (VIF) and tolerance statistics. The sociodemographic variables selected for entry into the model were: sex, age (under 65; 65 or above), education (under 12 years of education; 12 years of education or above), and living arrangements (either living alone or not). As for the clinical variables, the amount of time after diagnosis (3 years or less; more than 3 years); HbA1c (under 7; 7 or above); BMI (under 30; 30 or above); the care setting (primary care; hospital external consultations); having undertaken insulin treatment (no; yes); and the presence of associated complications (no; yes), as a proxy for severity, were included.

Data were analyzed using IBM SPSS (Statistical Package for the Social Sciences) version 28, and missing data were handled using the pairwise approach.

## 3. Results

### 3.1. Sample

A total of 763 persons were included in the final sample, where 51.5% were female. The youngest person was 24 and the oldest was 92 years of age, with a mean age of 66 ± 11.9 years. Table 1 shows the information regarding the sociodemographic and clinical characteristics of the observed sample. The majority of our sample was comprised of older individuals, individuals with lower education levels, and those living with someone else. 

In addition, it was observed that the most frequently associated pathology reported was hypertension (65.0%), followed by dyslipidemia (39.6%). Most of the reported complications caused by diabetes included retinopathy (19.3%) and arteriopathy (12.8%).

Table 2 presents the descriptive statistics for the DES-SF, DKT, and EQ-5D-5L; results for the DKT are separated by insulin treatment status, as explained in the methodology section, and the difference between the groups was statistically significant (|t| = 6.020, *p*-value < 0.001).

### 3.2. Determinants of Quality of Life

Spearman coefficients between EQ-5D-5L and both DES-SF and DKT were, respectively, 0.334 and 0.211; these are small values, but both were associated with a *p*-value < 0.001. The value for the correlation between DES-SF and DKT was 0.052 (*p*-value = 0.086).

The inferential analysis, shown in Table 3, showed that female patients had lower EQ-5D-5L scores, with a mean ± standard deviation of 0.60 ± 0.28 versus 0.67 ± 0.29 for male patients (*p*-value = 0.001). 

Patients who were 65 years or older had lower EQ-5D-5L scores compared to patients aged less than 65 years of age: 0.59 ± 0.30 versus 0.72 ± 0.26 (*p*-value < 0.001), respectively. As for living arrangements, patients living alone had lower EQ-5D-5L scores compared to those living with company: 0.59 ± 0.30 versus 0.65 ± 0.29 (*p*-value = 0.039). When comparing patients with different levels of education, patients with lower levels of education (i.e., under 12 years of education), had lower EQ-5D-5L scores compared to patients with 12 years of education or above (0.60 ± 0.29 vs. 0.78 ± 0.24; *p*-value < 0.001). Finally, patients with associated complications had lower scores compared to the ones with no complications (0.55 ± 0.29 vs. 0.70 ± 0.28; *p*-value < 0.001). All other sociodemographic and clinical values did not present significant values in univariate analysis. In addition, no significant associations were found between sex and sociodemographic and clinical characteristics.

The results for the linear regression model using the backward method are presented in Table 4 and show that sex, the presence of complications, age, and levels of HbA1c can be used to explain the EQ-5D-5L. The model showed no signs of violation regarding the linearity and normality of the residuals. The VIF factors ranged from 1.012 to 1.078, and the tolerance statistics ranged from 0.928 to 0.988. The Breusch–Pagan test for heteroscedasticity was not significant.

By analyzing the results for the EQ-5D-5L, the sociodemographic and clinical predictors involved in the model, it was observed that males presented higher EQ-5Q-5L scores, as did the younger age group and the group with no complications. The value of EQ-5D-5L increased as the DES increased, and similar results were observed in the analysis of the DKT variable.

## 4. Discussion

Studying QoL, as a fundamental feature of human life mediated by the value system, culture, goals, and people’s expectations [37], is essential for the implementation of interventions targeted at the specificities of each community.

The sample under study was mostly composed of women, although the prevalence of diabetes in Portugal is higher among males [38]. The present sample’s composition may be due to men being less likely to seek health surveillance appointments. In this sense, it is urgent that health professionals optimize health monitoring consultations of different scopes, paying particular attention to the underreporting of diabetes and, consequently, its complications [38,39], especially among men. Other research reflects some diversity regarding gender [40,41] in close articulation with the social construction of gender and its reflection on health. 

Similar to other studies, the sample was also characterized by having a high mean age (65.94 ± 11.86 years) [37], as well as a high number of people living alone and a high number of people with low academic achievement levels [40,41]. Thus, these people’s specific vulnerability to age-related comorbidities is enhanced by the high mean time to diagnosis, given the linear relationship between the duration of the disease course and the incidence of complications [39]. Considering diabetes is a condition that can be controlled but not cured, the essential aim of healthcare, particularly in older patients, is to delay its impairment in the remaining years of life and contribute to improvements in their QoL [40].

In this study, the results of the assessment of empowerment were moderate and the levels of knowledge were slightly lower among people undergoing insulin therapy, but much lower among people utilizing other therapeutic approaches. Some publications [42,43] have highlighted the relevance of knowledge about the disease in developing and improving the level of empowerment within people with diabetes, with a view toward changing behaviors, self-management, and self-control, as well as the improvement of their psychological status. Health services should devote greater attention to this finding because although chronic diseases, such as diabetes, require surveillance and monitoring by specialized health professionals, the effectiveness of their management especially depends on patients and their families [43].

Regarding QoL, the results obtained showed modest levels. QoL tends to be lower in people with chronic diseases [44], with diabetes being considered one of the most significant contributors to worse QoL levels [45]. Similar results have been evidenced in other studies, seeming to indicate, in general, that diabetes represents a psychological burden with the ability to negatively affect people in the emotional, physical, and social domains [41]. The disease challenges and the demands that treatment brings to the patients’ daily lives are identified as the main reasons for the decrease in perceived QoL [2,46].

Therefore, there is an emerging need, in addition to other indicators for monitoring the effectiveness of therapeutic treatments, to implement the systematic monitoring of QoL as a measure not only of the health status of people with diabetes but also of the implemented intervention [2]. It is about bringing in the voices of people with diabetes and looking at the impact of the disease and its treatment on physical, social, and mental well-being [13,47].

The data concerning the QoL assessment of people with type 2 diabetes and its association with sociodemographic factors revealed variability [37]. This study found statistically significant relationships between the QoL of people with type 2 diabetes and sociodemographic and clinical variables. In line with other studies, women, people aged 65 years or older, those with lower academic levels, and those who self-reported disease complications perceived worse QoL [37,41]. Statistically significant differences were observed related to BMI, HbA1c, time of diagnosis, setting of care, and insulin treatment. These results differ from other studies in which worse perceived QoL appeared to be associated with inverse scores of HbA1c [48], BMI [49,50], treatment using insulin [51] and the time of diagnosis, having considered 10 years as a cut-off point [9]. The fact that diabetes is a slow-evolving and silent disease, along with cultural aspects, may, at least in part, explain the relationships between perceptions of QoL and the clinical variables.

To identify statically significant factors associated with QoL in patients with type 2 diabetes, we built a multiple linear regression model. Thus, being younger, male, and not having diabetes complications were predictors of better perceived QoL.

The inverse relationship between age and QoL found in this study is corroborated by other research using similar statistical techniques [52]. However, other studies present contrary results, in which sociodemographic factors, such as age, gender, and education were not predictors of QoL [53]. In that study, the authors applied a specific instrument to measure QoL in people with type 2 diabetes, the AsianDQoL. This instrument has five domains: energy, memory, diet, sex, and finance, and the dimensions are not similar to those of the EQ-5D-5L, which could explain the contrary results.

A higher number of diabetes complications as a significant predictor of worse QoL is in line with other findings [1,52,54,55]. Other studies valuing complications focus on specific types of complications rather than the number, reporting that their severity is associated with worse QoL [53]. Donald et al. (2013) report that worse QoL related to diabetes complications is also associated with mental health conditions, such as depression and anxiety, and these relationships persisted after adjustment for sex, age, duration of diabetes, treatment regimen, and other clinical and sociodemographic variables [54].

Another result that should be highlighted is related to the observed BMI, which was substantially high in this study. A high BMI is usually significantly associated with lower perceived QoL [45], which was not observed in this study, even after adjusting for other covariates. This result contrasts with the study conducted by Timar et al. (2016), which found a significant association between obesity and QoL [51].

Despite the significant advances in the therapeutic approaches to diabetes mellitus, studies still point to the persistence of modifiable social determinants, gaps in glycaemic control, and the prevention of complications. Studies focusing on the knowledge and self-management skills of people with diabetes as determinants for achieving their goals, particularly regarding QoL are, however, less frequent [56].

Several studies have shown that empowerment can reduce HbA1c levels [43], trigger behavioral changes, and increase health knowledge, self-management and control [6], QoL, motivation, and resilience [10]. The published research has revealed the need for new empowerment practices that take the potential and constraints of the patients and their families as a starting point [57], as well as the beliefs of health professionals [42]. 

Empowerment programs should anticipate comprehensive and multifaced approaches [42] that are socially and culturally appropriate for the populations to function as partners in care [14]. Additionally, people with diabetes are adults—a substantial portion of whom are at advanced ages—whose lifestyles have been acquired and repeated over many years. Thus, behavioral changes are intended to be stable; imply complex psychological processes, in which there is a need to intentionally and consciously replace spontaneous behaviors with others that are healthier; and are therapeutically more effective. There is also the relevance of social interaction contexts, where inappropriate behaviors are often reinforced by family and friends and repercussions on health are not immediately visible. 

Approaches imminently oriented toward exclusively somatic goals and negotiating with people without considering cognitive specificities, such as memory and learning, as well as their social, familiar, affective, and academic contexts have proven to be ineffective.

Therefore, the role of healthcare professionals should focus on facilitating conditions that allow for the acquisition of in-depth knowledge about the disease and the strengthening of motivation and resilience so that people with type 2 diabetes can make decisions and successfully take responsibility for managing their disease.

It is urgent that the current models of care organization, which view individuals from an isolated perspective, be progressively replaced by other, collaborative models. These models look at people from all settings where they are found and use interactive strategies with intergenerational, peer, political, and community involvement. These models create new perspectives and opportunities to combat social and gender inequalities more effectively and are capable of positively influencing the modifiable social determinants of chronic diseases, particularly diabetes.

As a possible limitation, we may address that our sample was selected in an outpatient setting, so it does not include inpatients. Patients with more severe health conditions, including nephropathies, neuropathies, and amputations, are less prevalent in our sample. Another study, with a sample involving people with diabetes who are hospitalized or at home, could provide the inclusion of these complications in the statistical model that explains QoL. Additionally, this study did not address the impact of socioeconomic status on QoL, even though it is a determinant of health which determines diabetes. However, in Portugal, the impact may be mitigated because, for this chronic disease, the NHS/government supports about 100% of the expenses for insulin and antidiabetic drugs. Thus, even with the increase in costs due to diabetes in total health expenditure, the percentage of the financial burden for users has remained constant over time [58].

This study noted a high mean age of the participants (66 years), and more than 80% of them reported less than 12 years of education, which may reflect a lower level of health literacy. Future research with a younger or more educated sample may be relevant in order to compare results. It will also be interesting to apply other instruments to identify the perceptions, beliefs, and barriers of participants, as well as to measure self-efficacy, motivation, cultural safety, and others, with the aim of finding other determinants of QoL and supporting better strategies for the empowerment chronic disease self-management.

## 5. Conclusions

The general purpose of this study was to better understand the population of individuals with type 2 diabetes, particularly with regard to its more specific characteristics, namely knowledge and social impacts, in order to be able to build more specific and guided intervention programs for patients and their integration into the community.

The literature review indicates that diabetes is one of the chronic diseases with a higher impact on perceived QoL. In this study, QoL levels were moderate. Higher perceptions of QoL were evident in males, those of a younger age, those without complications, and those with greater knowledge of diabetes; these led to the empowerment of relevant predictors for a better perception of QoL.

Those with lower educational levels had lower perceptions of QoL; however, this effect was no longer observed when the model was adjusted for other factors. The values obtained for BMI, due to underlying health implications, are also worrying, despite apparently not having influenced perceptions of QoL.

This requires the adoption of systematic, diversified, and more effective prevention and treatment measures, such as those more culturally, socially, and affectively adjusted to the endogenous and exogenous resources of people with type 2 diabetes and their families. In addition, it requires a more proactive attitude from health systems regarding the postponement of healthcare among men.

In a new approach, the role of healthcare professionals should be to provide knowledge for people with diabetes to reflect on their disease and, mainly, improve communication processes so that informed decisions can be made in developing a self-management plan that fits each person realistically, clinically, socially, and psychologically.

## Figures and Tables

**Table 1 ijerph-20-04544-t001:** Sociodemographic and clinical characteristics of the observed sample.

Variable	Value	*n*	%
Sex	Female	393	51.5
	Male	370	48.5
Age (years)	<45	40	5.2
	45–65	248	32.5
	≥65	475	62.3
	Min–Max	24–92	
	Q1–Q3	59–74	
	Mean ± standard deviation	66 ± 11.9	
Education	Under 12 years	623	82.1
	12 years or above	136	17.9
Living arrangement	Living alone	89	11.8
	With company	665	88.2
Time after diagnosis (years)	≤3	190	25.2
	>3	565	74.8
	Min–Max	0–50	
	Q1–Q3	3–15	
	Mean ± standard deviation	10.8 ± 9.3	
HbA1c (%)	<7	254	45.9
	≥7	299	54.1
	Min–Max	4.9–13.6	
	Q1–Q3	6.2–8.2	
	Mean ± standard deviation	7.4 ± 1.5	
BMI (Kg/m^2^)	Normal weight (18.5–24.9)	119	16.1
	Overweight (25–29.9)	295	39.8
	Obesity (≥30)	328	44.2
	Min–Max	18.5–58.9	
	Q1–Q3	26.6–32.8	
	Mean ± standard deviation	29.9 ± 5.2	
Setting of care	Primary care	503	65.9
	Hospital external consultations	260	34.1
Insulin	Yes	264	34.6
	No	499	65.4
Presence of complications	Yes	252	40.3
	No	374	59.7

BMI, body mass index; HbA1c, glycated hemoglobin; Min, minimum; Max, maximum; Q1, first quartile; Q3, third quartile.

**Table 2 ijerph-20-04544-t002:** Descriptive statistics for the DES-SF, DKT, and EQ-5D-5L.

Variable	Value	Score
DES-SF	Min–Max	0–100
	Q1–Q3	56.25–81.25
	Mean ± standard deviation	67.89 ± 16.85
DKT (Insulin-treated)	Min–Max	0.17–0.96
	Q1–Q3	0.57–0.74
	Mean ± standard deviation	0.65 ± 0.14
DKT (Non-insulin-treated)	Min–Max	0.07–0.93
	Q1–Q3	0.50–0.71
	Mean ± standard deviation	0.58 ± 0.17
EQ-5D-5L	Min–Max	−0.38–1
	Q1–Q3	0.45–0.66
	Mean ± standard deviation	0.64 ± 0.29

Min, minimum; Max, maximum; Q1, first quartile; Q3, third quartile.

**Table 3 ijerph-20-04544-t003:** Comparison of EQ-5D-5L scores across different sociodemographic and clinical groups.

Variable	Value	M ± sd	|*t*|	*p*-Value
Sex	Female	0.60 ± 0.28	3.085	0.001 *
	Male	0.67 ± 0.29		
Age	Under 65	0.72 ± 0.26	6.187	<0.001 *
	65 or above	0.59 ± 0.30		
Living arrangement	Living alone	0.59 ± 0.30	1.759	0.039 *
	With company	0.65 ± 0.29		
Education	Under 12 years	0.60 ± 0.29	6.769	<0.001 *
	12 years or above	0.78 ± 0.24		
BMI	Underweight to overweight	0.64 ± 0.29	0.334	0.738
	Obesity	0.63 ± 0.29		
HbA1c	<7	0.61 ± 0.29	1.019	0.309
	≥7	0.64 ± 0.30		
Insulin	Yes	0.62 ± 0.29	0.817	0.414
	No	0.64 ± 0.29		
Time after diagnosis	≤3	0.66 ± 0.30	0.912	0.362
	>3	0.63 ± 0.29		
Setting of care	Primary care	0.63 ± 0.30	1.406	0.160
	Hospital external consultations	0.66 ± 0.27		
Presence of complications	No	0.70 ± 0.28	6.593	<0.001 *
	Yes	0.55 ± 0.29		

M ± sd, mean ± standard deviation; * one-tailed *p*-value of the independent *t*-test.

**Table 4 ijerph-20-04544-t004:** Linear regression model results for the EQ-5D-5L.

	Unstandardized Coefficients				95% Confidence Interval for β	
Predictors	β	Standard Error	t	*p*-Value	Lower Bound	Upper Bound
Constant	0.215	0.095	2.253	0.025	0.027	0.402
Sex	−0.105	0.027	−3.821	<0.001	−0.159	−0.051
Age	−0.118	0.028	−4.211	<0.001	−0.172	−0.063
HbA1c	0.053	0.030	1.767	0.078	−0.006	0.112
Presence of associated complications	−0.081	0.028	−2.916	0.004	−0.135	−0.026
DES	0.005	0.001	5.136	<0.001	0.003	0.007
DKT	0.319	0.104	3.074	0.002	0.115	0.523

Reference categories: Sex: Male; Age: under 65; HbA1c: under 7; Presence of complications: No. F(6,310) = 19.651, *p*-value < 0.001, R^2^ = 0.276 and R^2^ adjusted = 0.262; Breusch–Pagan test for heteroscedasticity: F(6,310) = 1.636, *p*-value = 0.137.

## Data Availability

The data that support the findings of this study are available from the corresponding author upon reasonable request.

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
