# Peer review of "Empowerment and Knowledge as Determinants for Quality of Life: A Contribution to a Better Type 2 Diabetes Self-Management"

_ijerph, 2023, doi:10.3390/ijerph20054544_

Round 1

Reviewer 1 Report

Thank you for the opportunity to review your work. I applaud the researchers and clinicians for their work with this patient population. The needs of people with chronic co-morbidities are of great concern to healthcare providers on every continent. I have specific recommendations that will strengthen the quality of your writing and this article.

Abstract

-       Patients with or above 65 years old have lower EQ-5D-5L scores, as well as patients who live alone, with an under secondary education and with presence of complications. Clarity is needed in the writing.

-diabetic people capacitating them to manage their condition

-       (people with diabetes not diabetic people) (enabling – not capacitating) Clarity is needed in the writing.

-This is inconsistent – what are you trying to say here?

Portugal recorded 9.9%, one of the highest prevalence rates in Europe. [5] Between 2009 and 2019, the age and sex standardized rate of avoidable hospital admissions for adults with diabetes decreased from 56 to 44 per 100,000 inhabitants, with 48 Portugal in second place after Iceland.

This does not make sense - what are you trying to say here?

- However, the percentage of financial burden for users has remained constant over time, and this increase in cost has not been accompanied by an increase in inequity. [8]

What are you saying with this paragraph:

In this context, clinical practices must assume an interactive and multidirectional approach, based on the values, principles, culture, and subjectivities, together with the anthropobiological and pathophysiological dimensions inherent to the process of care, in this case, people with diabetes and their families. More than adapting people to their disease, it is important to take people as care partners, to enhance their endogenous and exogenous resources, and to cope with the management of this chronic and complex disease. [13,14] It is necessary to (re)organize

-       Define: subjectivities? anthropobiological? pathophysiological dimensions? endogenous and exogenous resources? (re)organize?

-       Also in the former paragraph: (re)oriented? Why not simply organize and oriented?

-       This could be more straightforward with specific examples. Clarity is needed in the writing.

Again – what is meant by (co)construction? Why not simply state construction?

When you state: arising from the triptych of meanings inherent to people with diabetes:

-       Are you referring to the 3 panels of family, community, and health professionals? Or is this something else? Clarity is needed in the writing.

Methods:

-       followed by 139 the ambulatory of four health institutions

-       what is the Ambulatory?

“Under secondary” and “Secondary or above” do not translate to an international audience. Can you reframe this?

Were the females in your study similar to the males in your study (e.g., age, education, living arrangements, time after diagnosis, HbA1c, BMI, setting, Insulin, comorbidities)? If not, you cannot determine with significance that males with DM have superior QoL.

Author Response

Dear reviewer:

Our gratitude for your time and effort in evaluating our manuscript thoroughly. Your attention to detail and constructive feedback has been immensely helpful in improving the quality of our work.

Your insights and suggestions have been invaluable in strengthening our research and making it more robust. We have considered your comments and made necessary changes to our manuscript to ensure accuracy and clarity. We reinforce our appreciation for the opportunity to review our manuscript.

Once again, thank you for providing us with this opportunity to refine our work and for your dedication to advancing scientific knowledge in our field.

As we have received comments from six different reviewers, we had to deal with all of them and respect their viewpoints. We hope that our attempt to answer to all of them, did not result in any question no fully answered. If so, please let us know and will act accordingly.

We want to add that the manuscript has been subject to an English revision by an international translation company.

Best regards,

  1. Thank you for the opportunity to review your work. I applaud the researchers and clinicians for their work with this patient population. The needs of people with chronic co-morbidities are of great concern to healthcare providers on every continent. I have specific recommendations that will strengthen the quality of your writing and this article.

Thank you for your comment.

  1. (lines 19-20) “Patients with or above 65 years old have lower EQ-5D-5L scores, as well as patients who live alone, with an under secondary education and with presence of complications”. Clarity is needed in the writing.

We revised these two lines in the Abstract to meet your suggestion. To be clearer, we changed the reference to the ‘EQ-5D-5L’ to a more generic term as ‘quality of life’. In addition, we better specified what we mean by ‘under secondary education’ by replacing this term by ‘less than 12 years of education’.

Therefore, these two lines were rewritten as ‘Patients aged 65 years or older had lower QoL scores, as well as patients who live alone, who have less than 12 years of education and with presence of complications’.

  1. (lines 25-26). “literacy and empowerment are important to improve QoL of diabetic people capacitating them to manage their condition. (people with diabetes not diabetic people) (enabling – not capacitating) Clarity is needed in the writing.

This is a very useful point, and we fully agree with your assessment. Therefore, these lines were rewritten an ‘literacy and empowerment are important to improve QoL of people with diabetes, enabling them to manage their health condition’.

  1. This is inconsistent – what are you trying to say here? (lines 45-49). “Portugal recorded 9.9%, one of the highest prevalence rates in Europe. [5] Between 2009 and 2019, the age and sex standardized rate of avoidable hospital admissions for adults with diabetes decreased from 56 to 44 per 100,000 inhabitants, with Portugal in second place after Iceland”.

We greatly appreciate your constructive feedback and fully agree. We changed these sentences by the following: ‘Comparing to this European figure, Portugal showed 9.9%, one of the highest prevalence rates in Europe [5]. However, between 2009 and 2019, due to an evident better performance of primary health care, the age and sex standardized rate of avoidable hospital admissions for adults with diabetes was 56 per 100,000 inhabitants, a fourth best European score’.

  1. This does not make sense - what are you trying to say here? (line 71-72) “However, the percentage of financial burden for users has remained constant over time, and this increase in cost has not been accompanied by an increase in inequity. [8].

Thank you for your question. The sentence is correct, but we agree that without further explanation it may seem somehow contradictory with the previous one. This really happens (high health system expenditure and more or less the same patients’ financial burden) because this kind of chronic patients are protected by the NHS and have about 100% of government participation for their prescribed drugs. We have included this comment in the final discussion.

  1. What are you saying with this paragraph. (lines 89-96). “In this context, clinical practices must assume an interactive and multidirectional approach, based on the values, principles, culture, and subjectivities, together with the anthropobiological and pathophysiological dimensions inherent to the process of care, in this case, people with diabetes and their families. More than adapting people to their disease, it is important to take people as care partners, to enhance their endogenous and exogenous resources, and to cope with the management of this chronic and complex disease. [13,14] It is necessary to (re)organize

Define: subjectivities? anthropobiological? pathophysiological dimensions? endogenous and exogenous resources? (re)organize?

Also, in the former paragraph: (re)oriented? Why not simply organize and oriented? This could be more straightforward with specific examples. Clarity is needed in the writing.

All these concepts were clarified in the introduction. We maintained the word (re)oriented but we explained its utilization in the manuscript. In addition, we included some examples regarding the endogenous and exogenous resources. On the other hand, we decided to replace the words ‘subjectivities’, ‘anthropobiological’, ’pathophysiological’, ‘endogenous’ and ‘exogenous’ by other terms without need to be defined.

  1. Again – (line 104) what is meant by (co)construction? Why not simply state construction?

We fully agree with you and replaced the word ‘(co)construction’ by simply ‘construction’.

  1. When you state: (lines 112-115). ”Therefore, aiming at effective metabolic control, QoL, and subjective well-being [18], these conceptions confirm possibilities to combine personal and collective strategies [3], around new intelligibility, arising from the triptych of meanings inherent to people with diabetes, their family and community environment and health professionals”. Are you referring to the 3 panels of family, community, and health professionals? Or is this something else? Clarity is needed in the writing.

Once again, thank you for your comment. We rewrote the sentence as “Therefore, aiming at effective metabolic control, QoL, and subjective well-being [18], these conceptions confirm possibilities to combine personal and collective strategies [3], involving people with diabetes, their family and community environment and health professionals.”

  1. Methods: (139-140). “followed by the ambulatory of four health institutions”. what is the Ambulatory?

Regarding this point, we replaced the word ‘ambulatory’ by ‘external consultations’

  1. “Under secondary” and “Secondary or above” do not translate to an international audience. Can you reframe this?

This is a very useful point, and we fully agree with your assessment. The word ‘secondary’ may have different meanings across the countries. Therefore, we replaced it by the number of years of education. Then, through the manuscript, the term ‘under secondary’ was replaced by ‘with less than 12 years of education’ and the term ‘secondary or above’ was replaced by ‘with at least 12 years of education’

  1. Were the females in your study similar to the males in your study (e.g., age, education, living arrangements, time after diagnosis, HbA1c, BMI, setting, Insulin, comorbidities)? If not, you cannot determine with significance that males with DM have superior QoL.

We assessed the association between sex and sociodemographic and clinical characteristics. No significant association were found. This information was included in new lines 256 and 257.

Reviewer 2 Report

In the age group of those under 65 years of age, was it not contemplated to make several age groups? Younger people over 18 years of age are expected to have a better quality of life than those close to 65 years of age.

 Do you think that the socioeconomic level, as a determinant of health, could also have influenced the levels of quality of life?

Author Response

Dear reviewer:

Our gratitude for your time and effort in evaluating our manuscript thoroughly. Your attention to detail and constructive feedback has been immensely helpful in improving the quality of our work.

Your insights and suggestions have been invaluable in strengthening our research and making it more robust. We have considered your comments and made necessary changes to our manuscript to ensure accuracy and clarity. We reinforce our appreciation for the opportunity to review our manuscript.

Once again, thank you for providing us with this opportunity to refine our work and for your dedication to advancing scientific knowledge in our field.

As we have received comments from six different reviewers, we had to deal with all of them and respect their viewpoints. We hope that our attempt to answer to all of them, did not result in any question no fully answered. If so, please let us know and will act accordingly.

We want to add that the manuscript has been subject to an English revision by an international translation company.

Best regards,

  1. In the age group of those under 65 years of age, was it not contemplated to make several age groups? Younger people over 18 years of age are expected to have a better quality of life than those close to 65 years of age.

Thank you for your comment. As you know, type 2 diabetes is a disease that usually is detected in older individuals. In our sample, these categories were used because, as reported in table 1, 75% of patients have 59 or more and the median is 67 years. However, we included, in table 1, information regarding more detailed age groups for age under 65, for further understanding.

  1. Do you think that the socioeconomic level, as a determinant of health, could also have influenced the levels of quality of life?

This is a very useful point. Following your comment, we included in the limitation section of the discussion the following sentence: “This study did not address the impact of socioeconomic status on QoL, even though it is a determinant of health style which determines diabetes. However, in Portugal the impact may be mitigated because, for this chronic disease, the NHS/government supports about 100% of expenses for insulin and antidiabetic drugs”.

Reviewer 3 Report

Your study is to assess how knowledge and empowerment impact the QoL of a person with type 2 diabetes, leading to a better communication and disease management.  Despite your efforts, there are still some minor comments.

1) The Section Introduction is too long, I suggest that it should be rewritten and the contribution of the study should be clearly written.

2)In the manuscript, you need to elaborate the mechanism of the role of empowerment and knowledge as determinants for QoF in diabetes self-management.

3)You have neglected the role of wealth on the diabetes self-managemen, as well as the relationship with empowerment and knowledge of QoF. It is suggested that you supplement the relevant content too.

Author Response

Dear reviewer:

Our gratitude for your time and effort in evaluating our manuscript thoroughly. Your attention to detail and constructive feedback has been immensely helpful in improving the quality of our work.

Your insights and suggestions have been invaluable in strengthening our research and making it more robust. We have considered your comments and made necessary changes to our manuscript to ensure accuracy and clarity. We reinforce our appreciation for the opportunity to review our manuscript.

Once again, thank you for providing us with this opportunity to refine our work and for your dedication to advancing scientific knowledge in our field.

As we have received comments from six different reviewers, we had to deal with all of them and respect their viewpoints. We hope that our attempt to answer to all of them, did not result in any question no fully answered. If so, please let us know and will act accordingly.

We want to add that the manuscript has been subject to an English revision by an international translation company.

Best regards,

  1. Your study is to assess how knowledge and empowerment impact the QoL of a person with type 2 diabetes, leading to a better communication and disease management. Despite your efforts, there are still some minor comments.

Thank you for your comment.

  1. The Section Introduction is too long, I suggest that it should be rewritten and the contribution of the study should be clearly written.

Once again, thank you for your question. We have shortened the introduction. We hope it reaches your expectations.

  1. In the manuscript, you need to elaborate the mechanism of the role of empowerment and knowledge as determinants for QoL in diabetes self-management.

This is a very useful point, and we fully agree with your assessment. We rewrote the manuscript reinforcing the mechanism between empowerment and knowledge.

  1. You have neglected the role of wealth on the diabetes self-management, as well as the relationship with empowerment and knowledge of QoL. It is suggested that you supplement the relevant content too.

We greatly appreciate your constructive feedback and fully agree that we should have emphasized the role of wealth on the diabetes self-management, as well as the relationship with empowerment and knowledge of QoL. In the discussion we addressed this relationship, mentioning the financial protection people with diabetes have in Portugal, with the NHS/government practically assuming all the expenses in the acquisition of drugs, monitoring devices and any barriers to access to health care settings. Therefore, the wealth of people with diabetes does not impact significantly on the access to therapeutics or self-management.

Reviewer 4 Report

IJERPH 03.02.2023

Review:

Empowerment and knowledge as determinants for quality of life: a contribution to a better diabetes self-management

General feedback:

Reference section: please ensure the completeness of your references according to the stipulated journal format

Needs proofreading to make sure the language structure and grammar are adequately used. There are some major grammar issues throughout

There is room for improvement in the paper because the literature review needs to be extended and have more research findings discussed in it

By text:

Line 39-42: ….Recent epidemiologic data from 2021, suggest a global prevalence …” – missing word: please add diabetes? Also, please state the data were for all different types of diabetes or specific? Like T1DM, T1DM, GDM? The data were for adult populations or overall? Please state

Line   69-72: which country/countries those data represented? Please state specific

Line 80:  ….are fundamental. Please state fundamental for what?

Line 86: add word diabetic/diabetes – … effective diabetic/diabetes management

Line 101: please add citation/reference for WHO 2021

Line 134: four health institutions = please state their names. Please state reason(s) of selecting those 4?

Line 140: 85,820 elements = please state what elements stood for?

Did you assess validity of Diabetes Empowerment Scale Short-147 Form (DES-SF), Diabetes Knowledge Test (DKT) and EQ-5D-5L for your sample populations? Please state

Did you perform pilot study?

Line 256-267: do not include the result figures in the paragraph, as already presented in Table 3

Line 304-305, 314: what are your definitions for moderate, low, modest for: 1.” assessment of empowerment are moderate?”; 2. “levels of knowledge are slightly lower?” 3. “QoL, the results obtained showed a modest?” Any specific range of scoring to indicate moderate, low, modest?

Line 306: “Some publications [38]” – but only one citation?

Please shorten section 5. Conclusions (to answer your objectives). Move the remaining content to section 4. Discussion

Author Response

Dear reviewer:

Our gratitude for your time and effort in evaluating our manuscript thoroughly. Your attention to detail and constructive feedback has been immensely helpful in improving the quality of our work.

Your insights and suggestions have been invaluable in strengthening our research and making it more robust. We have considered your comments and made necessary changes to our manuscript to ensure accuracy and clarity. We reinforce our appreciation for the opportunity to review our manuscript.

Once again, thank you for providing us with this opportunity to refine our work and for your dedication to advancing scientific knowledge in our field.

As we have received comments from six different reviewers, we had to deal with all of them and respect their viewpoints. We hope that our attempt to answer to all of them, did not result in any question no fully answered. If so, please let us know and will act accordingly.

We want to add that the manuscript has been subject to an English revision by an international translation company.

Best regards,

  1. Reference section: please ensure the completeness of your references according to the stipulated journal format

Thank you for your comment. This has been done.

  1. Needs proofreading to make sure the language structure and grammar are adequately used. There are some major grammar issues throughout

Once again, thank you for your suggestion. The manuscript has been subject to an English revision by an international translation company.

  1. There is room for improvement in the paper because the literature review needs to be extended and have more research findings discussed in it

This is a very useful point, and we fully agree with your assessment. We have substantially changed the introduction.

  1. Line 39-42: …Recent epidemiologic data from 2021, suggest a global prevalence …” – missing word: please add diabetes? Also, please state the data were for all different types of diabetes or specific? Like T1DM, T1DM, GDM? The data were for adult populations or overall? Please state

Due to some reviewers’ comments, we had to drastically reduce the length of the introduction. Therefore, we had to delete some of the well-known epidemiological data.

  1. Line 69-72: which country/countries those data represented? Please state specific

From several other comments received we have also deleted this part of the manuscript.

  1. Line 80: …are fundamental. Please state fundamental for what?

At the end of this sentence, we have included the terms ‘type-2 diabetes self-care management’.

  1. Line 86: add word diabetic/diabetes – … effective diabetic/diabetes management

We have rephrased this sentience to the following: “However, effective diabetic management cannot be achieved only under the supervision of health professionals.”

  1. Line 101: please add citation/reference for WHO 2021

After the sentence “In 2021, WHO took the definition of empowerment and developed it further” we included the reference [15]”. However, the sentence was rephrased as 'In line with WHO, empowerment is characterized as personal and collective ability to contribute to the construction of policies and services, more oriented to their needs and potentialities'.

  1. Line 134: four health institutions = please state their names. Please state reason(s) of selecting those 4?

This is a very interesting point. The criterion to select these four institutions was because we wanted to have in our sample patients from external consultations of a large hospital and from consultations of health primary care centers. All these institutions belonged to the Northern Region of Portugal. They agreed with their participation in this study, but they did not formally authorize us to reveal the name of the institutions.

  1. Line 140: 85,820 elements = please state what elements stood for?

In this sentence the word ‘elements’ meant ‘the estimated total number of patients with diabetics seen by the four institutions’. We have included this clarification in the text.

  1. Did you assess validity of Diabetes Empowerment Scale Short-Form (DES-SF), Diabetes Knowledge Test (DKT) and EQ-5D-5L for your sample populations? Please state

This is a very useful point, and we fully agree with your assessment. However, the psychometric indicators from the Portuguese versions of these three measures are presented in the corresponding cited papers. All these papers have been mentioned in the methods section after presenting each measurement instrument.

  1. Did you perform pilot study?

Thank you for your question. As we had all main measurement instruments already validated for the Portuguese population, we did not perform any pilot test.

  1. Line 256-267: do not include the result figures in the paragraph, as already presented in Table 3

In the text we wanted to highlight the most relevant results from the table.

  1. Line 304-305, 314: what are your definitions for moderate, low, modest for: 1.” assessment of empowerment are moderate?”; 2. “levels of knowledge are slightly lower?” 3. “QoL, the results obtained showed a modest?” Any specific range of scoring to indicate moderate, low, modest?

You raise a very relevant point. However, the original authors of the measurement instruments for empowerment, knowledge and quality of life did not make any translation from numeric scores to a semantic meaning. This is the main reason why we did not want to include any other adjectives to our results. We only based our interpretation on the possible range of each scale.

  1. Line 306: “Some publications [38]” – but only one citation?

Thank you for your remark. We have mentions reference [38] just because it is already a literature review. However, to complement, we also included in the text the reference [39].

  1. Please shorten section 5. Conclusions (to answer your objectives). Move the remaining content to section 4. Discussion

Also, following other reviews we have changed these sections. Thank you.

Reviewer 5 Report

The study is of high interest and is very well designed and written. The results are clearly presented.

The limitations of the study should go in the discussion section, not in the conclusions. Another limitation may be the comparability of the data with other countries, given the social and cultural differences with Portugal. Although they mention suggestions for changes in the care model, they do not highlight guidelines for future research on the subject.

Other details:

1. Reference citations go before the point

2. Line 38: smoking is related to the effects and/or complications of diabetes, but not to the disease itself

3. Please separe the table 1 from the text (line 244)

4. Table 4 should go before the discussion

5. The titles of the tables should describe type of analysis, variables (dependent/independent, when corresponds), effect (associate, reduce, increase, relates...) and sample 

Author Response

Dear reviewer:

Our gratitude for your time and effort in evaluating our manuscript thoroughly. Your attention to detail and constructive feedback has been immensely helpful in improving the quality of our work.

Your insights and suggestions have been invaluable in strengthening our research and making it more robust. We have considered your comments and made necessary changes to our manuscript to ensure accuracy and clarity. We reinforce our appreciation for the opportunity to review our manuscript.

Once again, thank you for providing us with this opportunity to refine our work and for your dedication to advancing scientific knowledge in our field.

As we have received comments from six different reviewers, we had to deal with all of them and respect their viewpoints. We hope that our attempt to answer to all of them, did not result in any question no fully answered. If so, please let us know and will act accordingly.

We want to add that the manuscript has been subject to an English revision by an international translation company.

Best regards,

  1. The study is of high interest and is very well designed and written. The results are clearly presented.

Thank you for your comment.

  1. The limitations of the study should go in the discussion section, not in the conclusions.

Thank you for your remark. It is a very useful point. We have changed the manuscript accordingly.

  1. Another limitation may be the comparability of the data with other countries, given the social and cultural differences with Portugal. Although they mention suggestions for changes in the care model, they do not highlight guidelines for future research on the subject.

We greatly appreciate your constructive feedback and fully agree with it. At the end of discussion section, we have included the following guidelines for future research: ’Future research with a younger or more educated sample may be relevant in order to compare results. It will also be interesting to apply other instruments to identify perceptions, beliefs and barriers of the participants, as well as to measure self-efficacy, motivation, cultural safety and others, aiming to find other determinants of QoL and support the decision of better strategies to empower chronic diseases self-management.’

  1. Other details: Reference citations go before the point.

Thank you for your remark. We have made changes accordingly.

  1. Line 38: smoking is related to the effects and/or complications of diabetes, but not to the disease itself

We agree with you and deleted the reference to smoking. We also updated the reference to the OECDs Health at a Glance 2022.

  1. Please separate the table 1 from the text (line 244)

We added an extra space between the end of the table and next text. Thank you.

  1. Table 4 should go before the discussion

We completely agree with your comment. We moved tables 3 and 4.

  1. The titles of the tables should describe type of analysis, variables (dependent/independent, when corresponds), effect (associate, reduce, increase, relates...) and sample

Thank you for your comment. We included the suggestion on table 3 title, as all the others were already in the suggested format.

Reviewer 6 Report

Please see the attached document for comments/suggestions

Author Response

Dear reviewer:

Our gratitude for your time and effort in evaluating our manuscript thoroughly. Your attention to detail and constructive feedback has been immensely helpful in improving the quality of our work.

Your insights and suggestions have been invaluable in strengthening our research and making it more robust. We have considered your comments and made necessary changes to our manuscript to ensure accuracy and clarity. We reinforce our appreciation for the opportunity to review our manuscript.

Once again, thank you for providing us with this opportunity to refine our work and for your dedication to advancing scientific knowledge in our field.

As we have received comments from six different reviewers, we had to deal with all of them and respect their viewpoints. We hope that our attempt to answer to all of them, did not result in any question no fully answered. If so, please let us know and will act accordingly.

We want to add that the manuscript has been subject to an English revision by an international translation company.

Best regards,

  1. Title of the article - Empowerment and knowledge as determinants for quality of life: a contribution to a better diabetes self-management.
    From this statement “This chronic disease can be prevented by fighting the main risk factors, such as poor eating habits, physical inactivity, excessive weight, and smoking” it is likely that type 2 diabetes mellitus is the specific sub-type of diabetes that is the focus in this manuscript so please add type 2 in the title of the manuscript and consistently within the manuscript.

The title and the abstract were changed and ‘type 2’ was included. Same thing was performed along the manuscript.

  1. Significance - Chronic disease self-management [CDSM] has been increasingly recognised as helpful to individuals living with chronic disease(s) (See these articles as early exemplars
    - Barlow, J, Wright, C, Sheasby, J, Turner, A, Hainsworth, A. 2002. Self-management approaches for people with chronic conditions: a review, Patient Education and Counseling, 48(2); 177-187;
    - and also, Thorne SE, Paterson, BL. 2001. Health care professional support for self-care management in chronic illness: Insights from diabetes research, Patient Education and Counseling, 42 (1): 81-90.)
    - and implemented for approximately two decades (see here for an example Harris, MF, Williams, AM, Dennis, SM, Zwar, NA, and Powell Davies, G.2008. Chronic disease self-management: implementation with and within Australian general practice. Med J Aust; 189 (10): S17.)
    - (Just for context, reviewer has been teaching it to health students since 2000).

    Please see here for a discussion over development over time (Audulv, Å. The over time development of chronic illness self-management patterns: a longitudinal qualitative study. BMC Public Health 13, 452 (2013). https://doi.org/10.1186/1471-2458-13-452).
    In addition, the empowerment and sense of contributing to managing the condition has also been noted to develop in different patterns such as consistent, episodic, on demand or transitional (Audulv, Å. The over time development of chronic illness self-management patterns: a longitudinal qualitative study. BMC Public Health 13, 452 (2013). https://doi.org/10.1186/1471- 2458-13-452).
    Accordingly, there is an opportunity in this manuscript to strongly signpost the significance and implications of this study for CDSM, Type 2 diabetes, and Northern Portugal.
    Highlight please what this study adds to the existing body of knowledge, especially in areas such as quality of life which appears as a focus throughout the manuscript.

From this study we were able to obtain better knowledge from the community reality, and use it as a basis for further intervention aimed to improve the self-management of persons with diabetes. This study can also contribute to professional clinical practice, as we refer in introduction section ‘In this sense, it is important to proceed with the research supporting clinical practice on the QoL of people with type 2 diabetes and its relationship with empowerment and knowledge, with focus on more person- and family-centred care models.’

  1. Abstract - The abstract identifies the aims of the study, the type of study, some results and their implications.
    Consider modifying this sentence for consistency “It was also found that being male, having under 65 years age, have no presence of complications and higher values of knowledge and empowerment can predict higher QoL”. Perhaps to something like this “It was also found that being male, being under 65 years age, having no complications present, and, having higher values of knowledge and empowerment can predict higher QoL.”
    Additional clarity could be added to this statement, “Patients with or above 65 years old have lower EQ-5D-5L scores, as well as patients who live alone, with an under secondary education and with presence of complications.” This could be something such as, “Patients aged 65 years or older have lower EQ-5D-5L scores, as well as patients who live alone, who have completed less than secondary education and with presence of complications”.

We followed your suggestion for the first sentence. We rewrote the last one, and we also changed the reference to the ‘EQ-5D-5L’ to a more generic term as ‘QoL’, and ‘secondary education’ to ’12 years of education‘, for a better understanding: ‘Patients aged 65 years or older have lower QoL scores, as well as patients who live alone, who have less than 12 years of education and with presence of complications’.

  1. Key Words - Current key words are: QoL determinants; Diabetes; DES; DKT; EQ-5D-5. Since key words alert readers and researchers to topics of interest along with the abstract, consider whether DES is sufficiently targeted it is a widely used abbreviation for diethylstilboestrol in nursing, pharmacy, Dissociative experiences scale in psychology and psychiatry, drug-eluting stent in Cardiology and dry eye syndrome in Ophthalmology for example. There are 20+ acronyms listed here
    https://www.allacronyms.com/DES/medical.
    Consider adding words to supplement acronyms - something like ‘chronic disease self-management; disease state knowledge’ for additional clarity.

We changed key words to the following “Type 2 diabetes; Chronic disease self-management; Diabetes Empowerment Scale; Diabetes Knowledge Test”.

  1. Introduction - The Introduction briefly defines the disease state, provides some brief background to the issues and challenges associated with diabetes mellitus, focusses on Portugal and compares with other OECD nations. The need for and benefits from chronic disease self- management are outlined for people living with diabetes. The costs of poorly managed diabetes complications are also addressed to establish the gravity of the condition to the health system and the individual.

Thank you for your comment.

  1. The manuscript states, “Diabetes Mellitus is a metabolic disease characterized by hyperglycemia resulting from defects in insulin secretion and/or its action”. It is correct that Diabetes is a metabolic disorder, and, it is of multiple aetiology characterized by chronic hyperglycaemia with disturbances of carbohydrate, fat and protein metabolism resulting from defects in insulin secretion, insulin action, or both (American Diabetes Association (2013). Diagnosis and classification of diabetes mellitus. Diabetes care, 36 Suppl 1(Suppl 1), S67 S74. https://doi.org/10.2337/dc13- S067). These broad-ranging effects of diabetes mellitus include long term damage, dysfunction and failure of various organs necessitating self-management by the individual in addition to the various health professionals which may include general doctor, endocrinologist, podiatrist, optometrist or ophthalmologist, cardiologist, exercise physiologist, dietitian, pharmacist and dentist.

Thank you for your suggestion. We have included this reference in our text.

  1. The manuscript continues, “The Quality of Life of diabetic patients and the evolution of the disease depends on the ability for self-management. This requires higher patients’ knowledge about diabetes, enhancing the empowerment.”
    Does more knowledge automatically translate into empowerment and/or better chronic disease self-management?

We have included your concern in the introduction: ‘The management of diabetes implies an adequate level of knowledge, to develop critical awareness about the necessity for a change regarding the therapeutic process [10], promoting its effective adoption [4,6]. It requires not only the acquisition and deepening of specific knowledge in the area of diabetes, but also the ability of people to set their own goals and objectives’.

  1. The person living with diabetes needs to develop significant “disease competencies” which include using a glucometer, using a lancet device, understanding the readings and the trends including HbA1c, incorporating lower glycaemic index foods, examining the feet regularly, being adherent to medications, cutting down or ceasing smoking, very modest alcohol consumption if allowed, maintaining meticulous oral hygiene, using a medic alert bracelet/ necklet and understanding “sick day” management.

Thank you for your comment.

  1. Methodology - The study design, participants and study instruments were clearly stated, as was human ethics approval for the four health care recruitment sites, located in the north of Portugal. The inclusion criteria for participants were: diagnosis of type 2 diabetes over one year, enrolled in multidisciplinary consultation for at least three months, and older than 18 years of age.
    Please add the recruitment strategy, what information was provided to participants. And whether consent was written, verbal or implied. In addition, please describe how the study instruments were administered (privacy and confidentiality), where (private room, large auditorium for example) and by whom. How many/what percentage of potential participants approached to participate declined to do so?

We followed your suggestion and we added the sentences ‘The recruitment strategy was sequential. Each participant was guaranteed anonymity and confidentiality. A previous free informed written consent was always requested. The questionnaires were self-completed and privacy was safeguarded by using separate rooms. The rate of participation was around 90%’.

  1. The study instruments were comprised of Diabetes Empowerment Scale Short- Form (DES-SF), Diabetes Knowledge Test (DKT) and EQ-5D-5L, plus some sociodemographic variables (sex, age, education, and living alone) and clinical characteristics of the participants (body mass index - BMI, glycated haemoglobin HbA1c [proxy for diabetic control], treatment with insulin, and time of diabetes diagnosis).
    The statistical package (BM SPSS Statistical Package for the Social Sciences version 28) and analyses were stated, and any tests or models applied were identified.

Thank you for your remarks.

  1. Results/outcomes - The results section first characterizes the 763-participant sample using the sociodemographic and clinical focused data collected. This is presented in Table 1. Table 2 then presents the three test instruments and the participants scores, presenting DKT scores also by insulin included in management regimen or not.
    With respect to Quality of Life, Table 3 is presented, and it is reported that females presented with a lower score than males.

Thank you.

  1. Table 4 [Linear regression model results on EQ-5D-5L] is discussed in the Results at lines 269 and subsequent exploring results for the linear regression model using the backward method, however the Table is placed within the Discussion at lines 300 and subsequent. Please re-site to within the results section.

We greatly appreciate your constructive feedback and fully agree with it. We have already solved this issue.

  1. Discussion - The Discussion section notes that the study sample has a higher percentage of female participants even though type 2 diabetes is more prevalent in males proposing that males may be less likely to seek health monitoring appointments.
    Further it is observed that the sample mean is older, with more people living alone, with lower educational levels. This may reflect lower health literacy to be addressed as a potential intervention.

We included the following: ‘This study noted a high mean age of the participants (66 years), and more than 80% of them presented less than 12 years of education, which may reflect a lower level of health literacy. Future research with a younger or more educated sample may be relevant in order to compare results. It will also be interesting to apply other instruments to identify perceptions, beliefs and barriers of the participants, as well as to measure self-efficacy, motivation, cultural safety and others, aiming to find other determinants of QoL and support the decision of better strategies to empower chronic diseases self-management’.

  1. QOL results were identified as modest and consistent with other studies which proposed that diabetes is a condition that affects individuals emotional, physical and social domains, imposing many challenges and demands since individuals live with condition 24/7.The authors propose monitoring quality of life to evaluate these impacts, noting that it requires, “the systematic monitoring of QoL as a measure, not only of the health status of people with diabetes but also of the assessment of the implemented intervention”. Thus, quality of life changes would serve as a proxy measure for evaluation of an intervention which may perhaps benefit from other measures in addition - for example if the intervention were a change in medication, then measures of adherence might also provide insights.
    Further, the anticipation seems to be that measuring quality of life would provide information about the lived experience of type 2 diabetes, “… bringing in the voices of people with diabetes, about looking to the impact of the disease and its treatment on physical, social, and mental well-being”. It could enhance a reader’s understanding to tease out in a little more detail some of these links and how they articulate.

Thank you for your comment.

  1. Results consistent with some but inconsistent with others were noted but not explored, “The inverse relationship between age and QoL found in this study is corroborated by other research using similar statistical techniques. However, other studies present contrary results, in which sociodemographic factors, such as age, gender, and education were not predictors of QoL.”
    These results are interesting, were similar study instruments used? Were the participants living only with type 2 diabetes and not also other chronic conditions such as arthritis or asthma? Were there more similarities to the health systems of the studies that echo your results? Were some participants in the studies with contrary results perhaps immigrants or having different cultural practices such as valuing higher body weight?

We added the following: ‘In that study, the authors applied a specific instrument to measure QoL in people with type 2 diabetes, the AsianDQoL. This instrument has five domains: energy, memory, diet, sex and finance; and the dimensions are not similar with those from EQ-5D-5L, which can explain the contrary results’.

  1. It can be crucial to identify perceptions, beliefs, barriers or issues when proposing strategies to empower chronic disease self-management which is a focus of the manuscript.

Thank you for your comment.

  1. Conclusions - The Conclusion identifies the overall aim of the study as, “The general purpose of this study was to better understand the population of type 2 diabetes, particularly with regard to its more specific characteristics, namely knowledge and social impacts, in order to be able to build a more specific and guiding intervention programs for the patient and their integration into the community”.
    A recommendation arising from this study was, “These [intervention programs] requires the adoption of systematic, diversified, and more effective prevention and treatment measures, such as culturally, socially, and affectively more adjusted to the endogenous and exogenous resources of people with type 2 diabetes and their families. In addition, it requires a more proactive attitude from health systems, regarding the postponing of men in asking for healthcare”.

Thank you for your comment.

  1. The study results and acknowledged limitations, may not provide all the variables that a systematic program(s) needs to influence successful CDSM and higher quality of life. Factors such as health literacy in addition to knowledge, self-efficacy, health locus of control, motivation, insights from health belief models, cultural safety and others may hold the keys to success, so perhaps further studies with different or additional variables, instruments and participant groups might be required.

We have included your points in the further studies part of the discussion section.

  1. References - Fifty-three references cited are relevant to the research topics.

Thank you for your comment.

Round 2

Reviewer 1 Report

The authors have addressed my concerns and have greatly improved this manuscript. 

Reviewer 3 Report

Thank you for handling my comments. After your revision, I think it meets the requirements of IJERPH.